# The Mysterious Amurian Grig *Paracyphoderris erebeus* Storozhenko, 1980 (Orthoptera: Prophalangopsidae): New Data on Its Distribution, Ecology and Biology

**DOI:** 10.3390/insects14100789

**Published:** 2023-09-27

**Authors:** Sergey Yu. Storozhenko, Vladimir V. Molodtsov, Michael G. Sergeev

**Affiliations:** 1Federal Scientific Center of the East Asia Terrestrial Biodiversity, Far Eastern Branch of the Russian Academy of Sciences, 159 Avenue of the 100th Anniversary of Vladivostok, 690022 Vladivostok, Russia; 2Department of General Biology and Ecology, Novosibirsk State University, 1 Pirogova Street, 630090 Novosibirsk, Russia; vv@fen.nsu.ru (V.V.M.); mgs@fen.nsu.ru (M.G.S.); 3Laboratory of Biogeomodelling and Ecoinformatics, Novosibirsk State University, 1 Pirogova Street, 630090 Novosibirsk, Russia; 4Laboratory of Invertebrate Ecology, Institute of Systematics and Ecology of Animals, Siberian Branch, Russian Academy of Sciences, 11 Frunze Street, 630091 Novosibirsk, Russia

**Keywords:** Russian Far East, relict orthopteran, distribution, habitat, behaviour, biology, range, modelling, climate warming, conservation status

## Abstract

**Simple Summary:**

The rare relict species *Paracyphoderris erebeus* is endemic to the Russian Far East. The primary objectives of this study were (1) to provide the unpublished data on habitat, behaviour and biology of the Amurian grig, (2) to discuss some issues arising from long-term studies of distribution, and (3) to produce ecologo-geographic models of the species distribution over the Far East as the basis to forecast some possible shifts in its distribution. Ecologo-geographic modelling of the *P. erebeus* distribution in 2021–2060 shows that the species range will be relatively stable.

**Abstract:**

New data on distribution, ecology and biology of the rare extant species *Paracyphoderris erebeus* of the almost completely ancient family Prophalangopsidae (Orthoptera) are given. This montane species prefers humid areas with relatively low summer temperatures. Habits, mating behaviour and life history of *P. erebeus* are extremely similar to those of the North American representatives of the genus *Cyphoderris*. Nowadays, the Amurian grig is known from the Myaochan, Badzhalsky, Dusse-Alin, Bureinsky and Aezop ridges in the Khabarovsk Territory (north of the Amur River) and Bydyr Mountain in the Jewish Autonomous Region of Russia only. The analysis of the predicted distribution of *P. erebeus* based on the occurrence data reveals that the populations of the species may be more widely distributed over the southern part of the Russian Far East, at least up to 56° N. The ecologo-geographic model of the species distribution over its range is generated using the Maxent 3.4.4 software for the first time. Modelling of the *P. erebeus* distribution for 2021–2040 and 2041–2060 shows that the position of the species range appears to be relatively stable but a weak decline in the foretold suitability during climate warming may result in a reduction in population sizes and the subsequent fragmentation of the species population system. In this case, the Amurian grig will become a prospective to be included on the IUCN Red List.

## 1. Introduction

The Amurian grig, *Paracyphoderris erebeus* Storozhenko, 1980, was described in the family Haglidae from the Maochan Mountains near Komsomolsk-on-Amur, in the Khabarovsk Territory (Kray) of the Russian Far East [1]. Later, the genus *Paracyphoderris* Storozhenko, 1980, was transferred to the family Prophalangopsidae [2].

The family Prophalangopsidae is one of the few relict groups in Orthoptera known from the Early Jurassic to the present. This family possibly originated from the Triassic to Early Cretaceous family Haglidae, which is the ancestor of two large, recent superfamilies: Tettigonioidea and Stenopelmatoidea [3]. Present data show that Prophalangopsidae consists of about 100 species in 51 genera and seven subfamilies (the Jurassic Aboilinae, Chifengiinae, Protaboilinae, Termitidiinae, Tettohaglinae, the Jurassic to recent Prophalangopsinae, and extant Cyphoderrinae) [4]. As it contains the most fossil species and rare recent species, bearing some primeval characters, the Prophalangopsidae is interesting for studies on the evolutionary processes of Orthoptera and has attracted the attention of taxonomists who have worked on it [2,3,5,6,7,8,9,10,11].

The subfamily Prophalangopsinae contains three recent genera, *Prophalangopsis* F. Walker, *Tarragoilus* Gorochov and *Aboilomimus* Gorochov, and three Jurassic genera, *Jurassobatea* Zeuner, *Mesoprophalangopsis* Hong, and *Zalmonites* Handlirsch [4]. The genus *Prophalangopsis* was established in the family Gryllidae by F. Walker about 150 years ago based on a single male from “Hindostan” [12] (actually from the territory of the so-called British India—which is now several independent countries, including Bhutan, India, Myanmar and Nepal). The specimen is labelled with the very simple geographic label (“India”) without mentioning of an exact locality [13]. Later, this genus was transferred by W.F. Kirby [14] to its own subfamily Prophalangopsidae. Recently females of *P. obscura* (or similar species) were mentioned from Tibet [11]. Two genera, monotypic *Tarragoilus* and *Aboilomimus* (with two species), are distributed in the Hengduan Mts. and Yunnan–Guizhou (Yungui) Plateau (China: Sichuan and Guizhou) [9,11]. Bionomics of all extant forms of the subfamily are almost unknown.

The extant subfamily Cyphoderrinae consists of four species in two genera. The genus *Cyphoderris* Uhler was created by P.R. Uhler [15] for *C. monstrosa* Uhler from Oregon, USA. Later, two species and one subspecies of this genus were described from North America [16,17,18]. The biology of *Cyphoderris* is quite well known and some species of the genus are often used in studies on mating behaviour and stridulating. The stridulatory apparatus of males is almost perfectly symmetrical, that is, males have fully functional files and scrapers on both tegmina and can produce sound with either the left or the right tegmen [19]. The male hindwings of *Cyphoderris* are reduced to fleshy lobes, which are devoured by the female during copulation. Moreover, the Ander’s organ is present in both sexes at all life stages and works as an anti-predator ultrasound mechanism [20]. This organ is a unique stridulatory structure in the subfamily Cyphoderrinae compared to other Orthoptera. However, the Russian representative of Cyphoderrinae is poorly studied compared to the American species. This is why orthopterists call *Paracyphoderris erebeus* the enigmatic species [21].

Usually, the Amurian grig is discussed in studies on morphology, taxonomy and evolution of Orthoptera [2,3,10,22], while data on its distribution are limited by references to the Khabarovsk Territory of Russia only [23,24]. The current data on a few localities of this species on the Badzhalsky, Dusse-Alin and Aezop ridges are provided in two papers only [25,26]. Data on biology and ecology of *P. erebeus* are almost completely absent. Herein, we summarize all available data on habits, habitat and life history of this relict species and generate an ecologo-geographic model of the species distribution over its range using the Maxent 3.4.4 software for the first time.

The aims of this paper are (1) to provide new data on the habitat, behaviour and biology of *P. erebeus*, (2) to discuss some issues arising from long-term studies of distribution, and (3) to produce ecologo-geographic models of the species’ distribution across the Far East as the basis to forecast some possible shifts in its distribution.

## 2. Materials and Methods

### 2.1. Study Territory

Original data were collected from 1976 until 2022 in the southern part of the Khabarovsk Territory (Kray) and in the Jewish Autonomous Region (Oblast′). There are numerous relatively short and low (up to 2263 m) mountain ridges and local plains with rivers and lakes. This area is covered mainly by coniferous (taiga) and mixed forests. The upper parts of local ridges are often above the timberline. Ecosystems in its southern part are often converted to clearings, openings, agricultural lands (fields and pastures) and urban territories. However, there are several natural reserves (Bastak, Bolon, Bureya and Komsomolsk) as well. Across the local plains, average temperatures are relatively moderate (mean temperatures of the warmest month are between 16.5 °C and 21.5 °C, the same for the coldest month—from –19 °C to –26 °C), and annual precipitation amounts between 570 and 710 mm [27].

### 2.2. Original Data

We analysed all data on specimens collected by Russian researchers and all applicable published data [1,25,26]. Totally, 69 specimens of *Paracyphoderris erebeus* are examined. The holotype and paratypes were collected in the Khabarovsk Territory (Myaochan Ridge, 40 km NW of Komsomolsk-on-Amur, settlement Tikhy, 50.7458° N, 136.5114° E, 650 m, 23.08.1976, 1♂, 2♀, coll. V.A. Mutin). All other localities are listed in the Appendix A. Almost all specimens are kept in the Center of Biodiversity FEB RAS (Vladivostok) and the only holotype and paratypes of *P. erebeus* are stored in the Zoological Institute, Russian Academy of Sciences (Saint Petersburg, Russia). We mainly used Google Earth Pro (©Google, 2022) to determine geographical coordinates of localities.

### 2.3. Photographs

Photographs of the specimens fixed in alcohol were taken with an Olympus SZX16 stereomicroscope and an Olympus DP74 digital camera, and then stacked using Helicon Focus software (version 8.2.2). The final illustrations were post-processed for contrast and brightness using Adobe^®^ Photoshop^®^ version 7.0 software.

### 2.4. Data Analysis

Maps of species distribution were produced on the basis of geographic coordinates with MapInfo 15.2.4 (© Pitney Bowes Software Inc., Lanham, MD, USA; now, © Precisely, Burlington, MA, USA). A Lambert conformal conic projection was used as the basic map.

We employed the Maxent 3.4.4 software [28,29,30,31] to model the species distribution across the region. We preferred this software because it is very standardized [30] and the interface is relatively user friendly. For ecomodelling, we used data from WorldClim 2 [32,33] such as “Historical climate data” (19 standard annually averaged bioclimatic variables at the 30 arcsecond spatial resolution) and “Future climate data” (19 standard averaged bioclimatic variables) for 2021–2040 and 2041–2060 downscaled from the global climate models [33], namely CNRM-ESM2-1 (Centre National de Recherches Météorologiques and Centre Européen de Recherche et de Formation Avancée en Calcul Scientifique, France) [34,35] and for the three Shared Socioeconomic Pathways (1–2.6, 2–4.5, 3–7.0) [36].

Both the ecomodelling method and the data on climatic parameters have some limitations. The MaxEnt models are based only on occurrence data and depend on the number of occurrences, selected characteristics of modelling and selected sets of variables [29,30,31]. These limitations become very important for an analysis of small samples [37]. The WorldClim datasets include spatially interpolated climatic data, and their actual reliability is partly associated with densities of weather stations (relatively low for the region studied) [32]. In any case, we used the full sets of applicable bioclimatic variables to compare results for the same territory, but for different periods and climatic models. We rated accuracy of our models by using the AUC (the area under the receiver operating characteristic curve) values for sets of 16 replicates with cross-validation, and estimated the significance of climatic variables by their predictive contributions and Jackknife tests. We produced the models with the following parameters: features—auto; output format—cloglog [29]; and regularization multiplier = 1.

## 3. Results

### 3.1. Habits and Habitats

The known distribution of the Amurian grig is limited by the mountain areas of the southern part of the Khabarovsk Territory (north of the Amur River) and the Jewish Autonomous Region (Figure 1). *Paracyphoderris erebeus* is a montane species and typically inhabits the subalpine zone at an elevation of 500–1100 m in elfin woodland (*Pinus pumila*) and in larch forests (*Larix gmelini*) or, at a higher elevation (1200–1650 m), it occupies the tundra altitudinal belt, which is nearly devoid of vegetation. The preferable habitats are open rocky places covered mainly by the reindeer lichen (*Cladonia rangiferina*).

Usually, females and nymphs are found under stones where they construct borrows 2–3 cm in diameter but, during evenings, both can be observed in open places. Males are more active, and in sunny weather, they are found on the soil surface, on the reindeer lichen and on trees.

During the spring, *P. erebeus* feeds on the blossom of understory plants such as the lingonberry (*Vaccinium vitis-idaea*). In June, both adults and nymphs were observed on the elfin trees, feeding upon staminate cones prior to their ‘loose pollen’ stage (Figure 2).

In Prophalangopsidae, like in other katydids (or bush-crickets, Orthoptera: Ensifera), pure-tone and broadband sound production has evolved as a key mechanism for mate attraction and conspecific recognition. These sounds are produced by tegminal stridulation—the process of moving a hardened scraper on one forewing (tegmen), against a row of teeth (the file) on the other, producing vibrations on the wing, which are then amplified by specialized wing cells to radiate sound [13]. The American *Cyphoderris monstrosa* is unusual, because it can switch the position of the tegmina during stridulation, so that first one, then the other, is on top. Unlike *C. monstrosa*, a change in tegmen overlap is very rare in *P. erebeus*.

The stridulating males of *P. erebeus* were observed on sunny days on tree trunks at a height of about 30–40 cm above the soil level (Figure 3) or on ground covered by reindeer lichen (Figure 4). At this point, it is similar to *Cyphoderris buckelli* Hebard and *C. strepitans* Morris et Gwynne and differs from *C. monstrosa*. The stridulating males of the latter species occur on trees, often 5 m or more above ground level [18].

Like other Cyphoderrinae, the adults and nymphs of *P. erebeus* have the Ander’s organ and use it as an anti-predator ultrasound mechanism [20]. Undoubtedly, this unique morphological structure may be used as the diagnostic character of the subfamily Cyphoderrinae.

The mating behaviour of *P. erebeus* is extremely similar to that of *C. monstrosa*. The male hindwings are diminished to fleshy lobes (Figure 5), which are consumed by the female during copulation. Males who had already mated once and are missing these courtship “snacks” must resort to other methods of holding the female’s attention and instead use the “gin trap”, a complex system of cuticular modifications whose role is to hold the female’s abdomen firmly in place during copulation. The resting of such fixations is observed on the ventral side of the female’s abdomen as dark spots (Figure 6).

It seems that this species oviposits on the ground surface because its ovipositor is very short, almost rudimental (Figure 6). Probably to protect the eggs, this process takes place in cameras prepared by the female.

### 3.2. Life History

Under natural conditions, the adults of *P. erebeus* occur in June and remain active until the onset of the cold weather. This species overwinters as nymphs. To confirm this fact, the ten last instar nymphs were collected on the Myaochan Ridge on 29 August 1983 by the first coauthor (S.S.) and transported to Vladivostok. At the end of September of 1983, nymphs were placed in glass jars full of reindeer lichen. These jars were buried in the ground in a broad-leaved forest and covered by leaf litter. While the common temperature in Vladivostok is about –20 °C in January and February, all nymphs were found alive in April of 1984.

### 3.3. Ecological Models of the Species Distribution

The analysis of the predicted distribution of the Amurian grig based on the occurrence data (Figure 7) reveals that the populations of this species may be distributed over the southern part of the Russian Far East more widely, at least up to 56° N. The optimal habitats with a level of suitability more than 0.6 are in the central parts of the Khabarovsk Territory, in the northern parts of the Amur Region and in the southeastern parts of the Republic of Sakha (Yakuitia), mainly along the Stanovoy (Outer Khingan), Dzhagdy and Tukuringra Ranges. Unfortunately, this area is poorly studied.

The optimal habitats for this species are represented in the northern part of Sikhote-Alin separated from the known species range by the Amur River, near the southwestern shore of Lake Baikal (Khamar-Daban and Tunka Goltsy Ranges), and in the Changbai and Hamgyong Mts. The last two areas are far away from the known range. If the evident low mobility of *P. erebeus* is considered, there is a low probability of its occurrence in these regions, however, new species from the genus *Paracyphoderris* may be found in some places.

The model performance is very high, because the AUC value for 16 replicates equals 0.998 (Figure 8). Precipitation in the warmest quarter, annual precipitation, temperature annual range, temperature seasonality and mean temperature in driest quarter are the most important variables (Table 1). The Jackknife test shows a similar distribution of variable significance, but allows us to add two important characteristics (Figure 9), namely precipitation in the wettest quarter and in wettest month. This means that the species prefers more or less humid areas with relatively low summer temperatures.

Ecological modelling of the *P. erebeus* distribution in 2021–2040 and 2041–2060 based on the CNRM-ESM2-1 climatic model for three Shared Socioeconomic Pathways (1–2.6, 2–4.5, 3–7.0) shows that the local parts of the range may slightly shift northwards and north-westwards (Figure 10). In all cases, whatever the scenario (associated with either low or high emissions) and period (either 2021–2040 or 2041–2060) chosen, the predictions look very similar. In the main area, where the species is distributed now, the level of suitability of conditions will slightly decrease. However, in the northern parts (mainly along the Stanovoy Ridge), the suitability will become better. In all other areas (near Baikal Lake, in the northern parts of Sikhote-Alin Range, and near the border between China and North Korea), the suitability of conditions will decline as well.

Hence, the predicted patterns of the possible distribution of *P. erebeus* in the future reveal a very weak trend of northward shifts. The position of the known species range looks likely to be relatively stable until 2060. However, a weak decline in the foretold suitability may result in a reduction in population sizes and a subsequent fragmentation of the species population system.

## 4. Discussion

### 4.1. Patterns of Spatial Distribution of the Modern Taxa of Prophalangopsidae

In the Jurassic and Lower Cretaceous, prophalangopsids became one of the very widely distributed and abundant groups of insects [6,39]. They were one of most prominent components of the so-called soundscapes [38]. However, later their diversity and abundance considerably decreased. Now, we can find only members of some extant genera and species. All of them are distributed within the Northern Hemisphere, either in Asia (its eastern and southeastern parts) or in the western parts of North America. Unfortunately, the exact range of *Prophalangopsis obscura* remains unknown, but it may be distributed over the Eastern Himalayas, south-east Tibet and/or the Hengduan Mts., from where two other genera with three species were described and where two potential females of this (or similar) species were collected [11,13].

All other modern genera of the subfamily Prophalangopsinae occur in the subtropical mountains of China (Figure 11), mainly at middle and high altitudes (from 900 to 4200 m—*Tarragoilus diuturnus* Gorochov [9,11,40]). The last species was found under stones near streams in the mixed forest altitudinal belt [40]. Unfortunately, we know almost nothing about bionomics of the recent prophalangopsins. All known forms are associated with the Hengduanshan Province and probably with the East Himalayan and Tibetan Provinces, i.e., areas with an extremely complicated relief and with a very high diversity of Orthoptera [41,42].

Distribution of the subfamily Cyphoderrinae looks quite different. Both genera occur in the boreal and sub-boreal regions, either in eastern Asia or in western North America (Figure 11). Populations of all known species occur in the coniferous/mixed forest altitudinal belts, mainly the subalpine ones, and either in mountain tundras (*P. erebeus*) or in high altitude sagebrush prairies (*C. strepitans*) [18,43]. The evolution of both branches of the subfamily (Asian and American) may be explained by their strict associations with boreal, relatively cold and wet, types of ecosystems, probably originating at the end of the Pliocene [44,45].

### 4.2. Bionomics of Cyphoderrinae

Our knowledge about the bionomics of *P. erebeus* is limited (Section 3.1 and Section 3.2). However, many peculiarities of its life look similar to those of the well-known populations of its American relatives.

All Cyphoderrinae are adapted to relatively cold conditions. Their life cycles include overwintering nymphs and adults emerging at the end of spring or the beginning of summer. During inactive times (especially during dormancy), both nymphs and adults use some kind of shelter (under stones, in burrows, etc.). Lethal temperatures for *C. monstrosa* are −9–12 °C [46]. Its nymphs can be active at around 0 °C temperatures [46]. Moreover, males of *C. strepitans* may stridulate at air temperatures of around 0 degrees C [18].

There are some differences in diurnal activity, because the Amurian grig is characterized by diurnal activity, but the species from North America are active during nights. All species feed mainly on flowers of different shrubs, but *C. monstrosa* and *P. erebeus* also use staminate cones of pines (*C. monstrosa*—*Pinus contorta* [18] and *P. erebeus*—*Pinus pumila* (Section 3.1)) (Figure 2). Behaviour of *C. monstrosa* males is characterized by evident territoriality and escalated aggression [47], but the males of *C. strepitans* do not show territorial and aggressive attitudes [43]. The males of *P. erebeus* never demonstrate aggressive behaviour as well.

### 4.3. Consevation Biology of the Amurian Grig

The whole range of *Paracyphoderris erebeus* occupies about 65,000–70,000 km^2^ (Figure 1). The predicted area of its possible distribution covers more than 150,000 km^2^ (Figure 7). There are at least 16 localities where the species occurs. Taking into account the explicitly low mobility of these insects, the numbers of its local populations can be estimated to be at least 10–11. However, we have only some very restricted data on the species abundance. This is why the species may be characterized as Data Deficient (DD) [48]. In the future, global warming will result in some reductions of population sizes and fragmentation of the species population system [49]. In this case, the IUCN Red List category might change. Now, *P. erebeus* is not included in the Federal and regional Red Books, but its range covers territories of the two State Nature Reserves, namely Bureinsky and Bastak [26].

## 5. Conclusions

The Amurian grig is a rare relict species of Orthoptera and may be called a ‘living fossil’. This species prefers more or less humid areas with relatively low summer temperatures. The preferable habitats are open rocky places covered mainly by the reindeer lichen at elevations between 500 and 1650 m. Habits, mating behaviour and life history of *Paracyphoderris erebeus* are extremely similar to those of the North American representatives of the genus *Cyphoderris*. The known distribution of *Paracyphoderris erebeus* is limited by the mountain areas of the southern part of the Khabarovsk Territory (northward the Amur River) and the northern part of the Jewish Autonomous Region. The analysis of the predicted distribution of the Amurian grig based on the occurrence data reveals that the populations of the species may be distributed over the southern part of the Russian Far East more widely, at least up to 56° N. Ecological modelling of the *Paracyphoderris erebeus* distribution in 2021–2040 and 2041–2060 shows that the position of the species range looks relatively stable but a weak decline in the foretold suitability during climate warming may result in a reduction in population sizes and subsequent fragmentation of the species population system. In this case, the Amurian grig may be included on the IUCN Red List.

## Figures and Tables

**Figure 1 insects-14-00789-f001:**
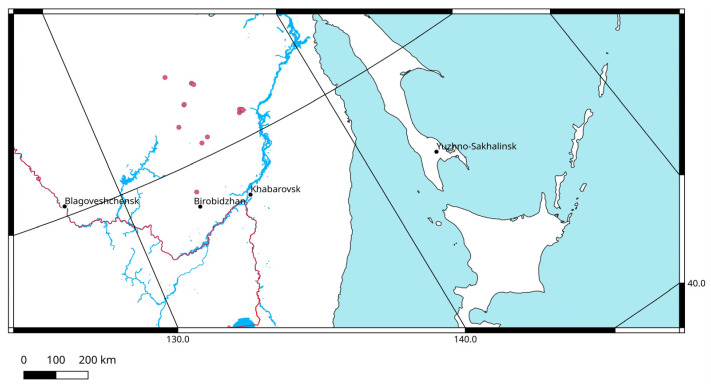
Distribution of *Paracyphoderris erebeus*.

**Figure 2 insects-14-00789-f002:**
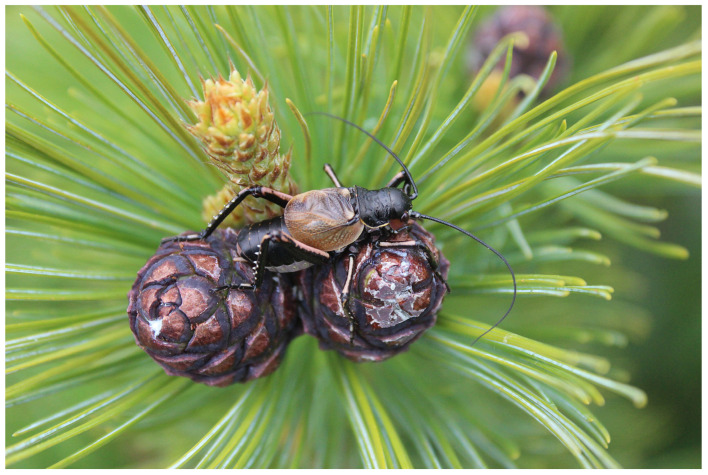
Male *Paracyphoderris erebeus* on an elfin tree (Myaochan Ridge, June 2023). Photograph by E. Krutikova; reproduced with author’s permission.

**Figure 3 insects-14-00789-f003:**
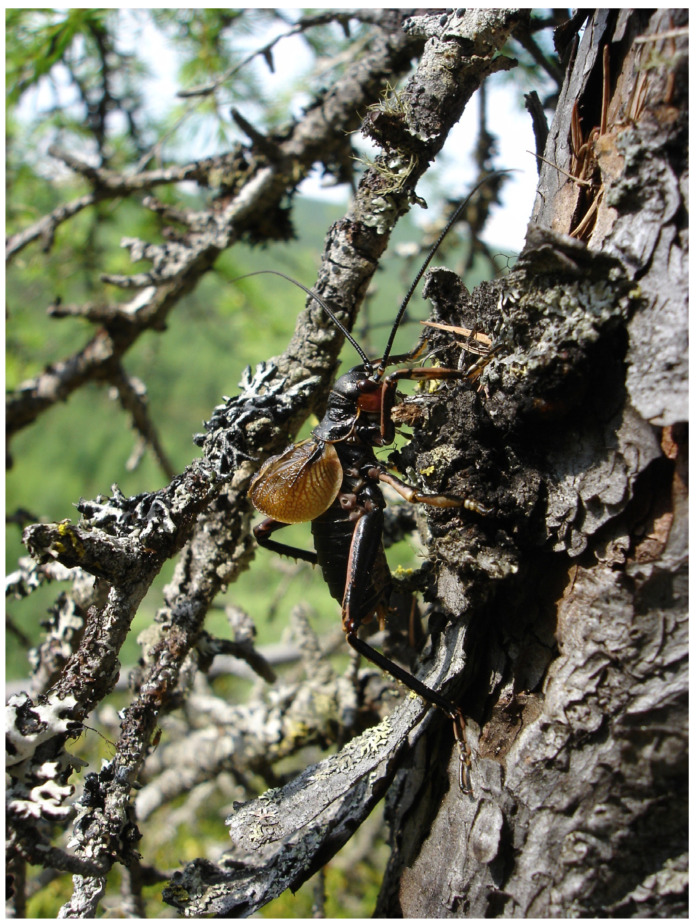
Stridulating male *Paracyphoderris erebeus* on the larch trunk (Dusse-Alin Ridge, June 2009). Photograph by E. Koshkin; reproduced with author’s permission.

**Figure 4 insects-14-00789-f004:**
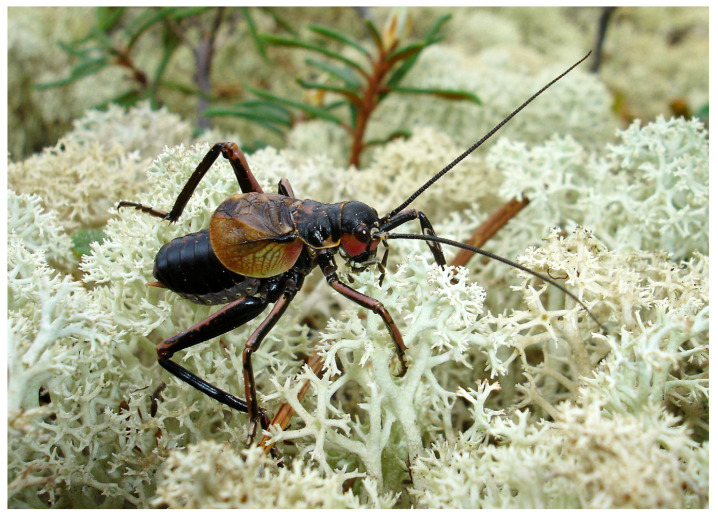
Stridulating male *Paracyphoderris erebeus* on the reindeer lichen (Dusse-Alin Ridge, June 2009). Photograph by E. Koshkin; reproduced with author’s permission.

**Figure 5 insects-14-00789-f005:**
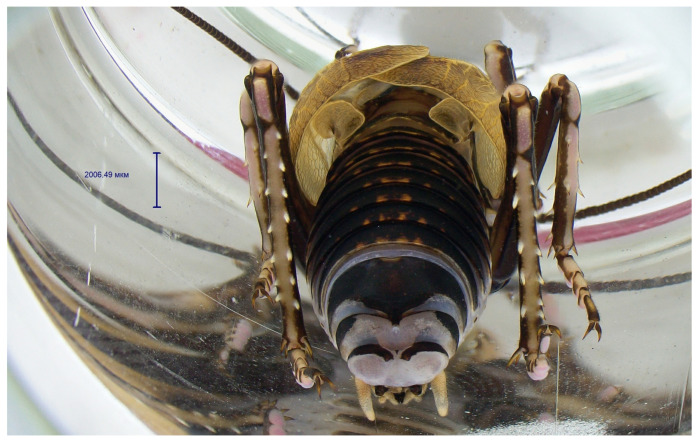
The body of *Paracyphoderris erebeus* male fixed in alcohol, dorso-posterior view.

**Figure 6 insects-14-00789-f006:**
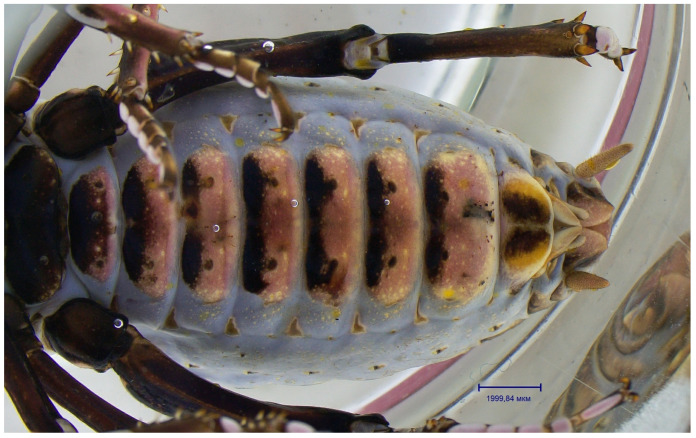
The abdomen of *Paracyphoderris erebeus* female fixed in alcohol, ventral view.

**Figure 7 insects-14-00789-f007:**
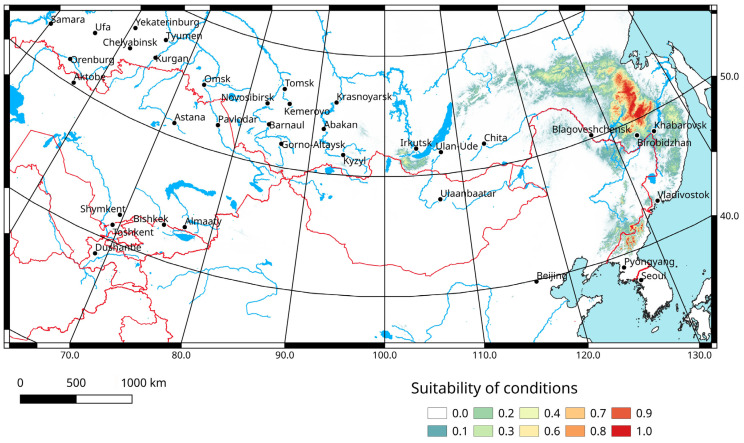
Predicted probabilities of suitable conditions for *Paracyphoderris erebeus* (bioclimatic variables for 1970–2000; point-wise mean for 16 replicates).

**Figure 8 insects-14-00789-f008:**
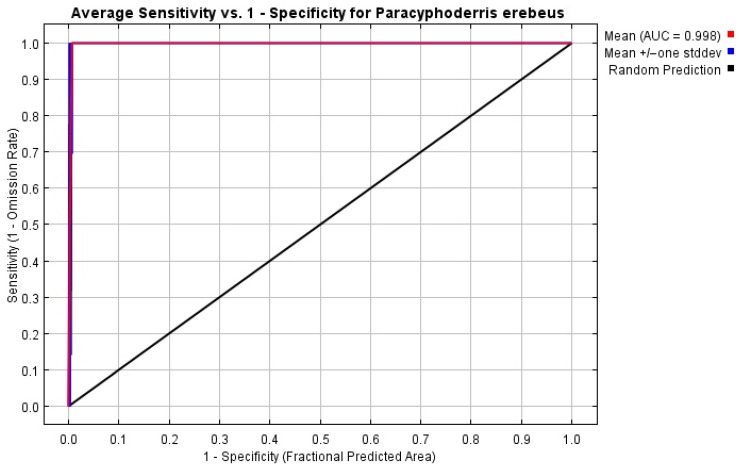
Reliability test for the distribution models of *Paracyphoderris erebeus* (bioclimatic variables for 1970–2000; 16 replicates with cross-validation).

**Figure 9 insects-14-00789-f009:**
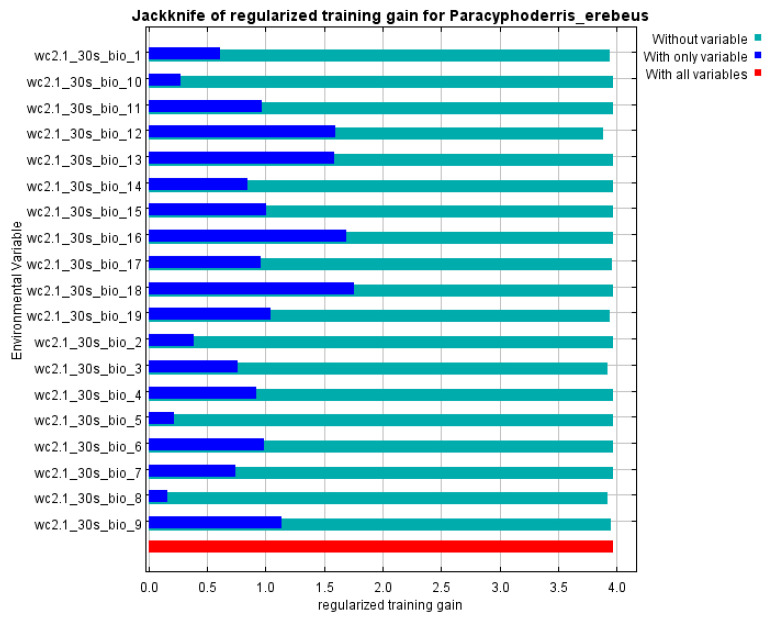
Jackknife of regularized training gain for the distribution models of *Paracyphoderris erebeus* (bioclimatic variables for 1970–2000; 16 replicates with cross-validation).

**Figure 10 insects-14-00789-f010:**
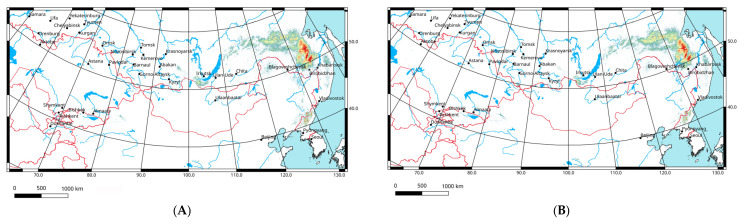
Predicted probabilities of suitable conditions for *Paracyphoderris erebeus* (all distribution data; forecasts of bioclimatic variables for 2021–2040 and 2041–2060 according the global climate model CNRM-ESM2-1 [36,37]; point-wise mean for 16 replicates): (**A**,**C**,**E**)—2021–2040; (**B**,**D**,**F**)—2041–2060; (**A**,**B**)—the 1–2.6 Shared Socioeconomic Pathway based on low greenhouse gas emissions; (**C**,**D**)—the 2–4.5 Shared Socioeconomic Pathway based on intermediate greenhouse gas emissions; and (**E**,**F**)—the 3–7.0 Shared Socioeconomic Pathway based on high greenhouse gas emissions [38].

**Figure 11 insects-14-00789-f011:**
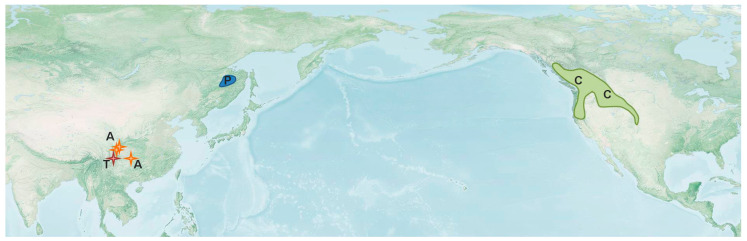
Distribution of the extant genera of the family Prophalangopsidae (except *Prophalanhopsis*): *Cyphoderris* (C); *Paracyphoderris* (P); *Aboilomimus* (A); and *Tarragoilus* (T). The basic map is based on “The NASA explorer base map”. The source image is a product of NASA Visible Earth Project.

**Table 1 insects-14-00789-t001:** Predictive contributions for all data.

Variable	Variable Explanation	Percent Contribution	Permutation Importance
bio_1	annual mean temperature	0.6	15.6
bio_2	mean diurnal range (mean of monthly (max temp—min temp))	0.3	0
bio_3	isothermality (bio2/bio7) (×100)	1.3	3.6
bio_4	temperature seasonality (standard deviation ×100)	7.1	0.2
bio_5	max temperature of warmest month	0.1	0
bio_6	min temperature of coldest month	0.4	0
bio_7	temperature annual range (bio5-bio6)	9.7	0
bio_8	mean temperature of wettest quarter	3.3	3.9
bio_9	mean temperature of driest quarter	7.0	10.1
bio_10	mean temperature of warmest quarter	0	0
bio_11	mean temperature of coldest quarter	1.6	0
bio_12	annual precipitation	17.4	28.4
bio_13	precipitation of wettest month	0	0
bio_14	precipitation of driest month	0.2	0.2
bio_15	precipitation seasonality (coefficient of variation)	1.9	0
bio_16	precipitation of wettest quarter	3.9	0
bio_17	precipitation of driest quarter	3.5	2.9
bio_18	precipitation of warmest quarter	40.6	0
bio_19	precipitation of coldest quarter	1.2	35.0

Highlighted in green—five most significant variables.

## Data Availability

Supplementary information or data can be provided upon request from the corresponding author.

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
