# Peer review of "The Mysterious Amurian Grig Paracyphoderris erebeus Storozhenko, 1980 (Orthoptera: Prophalangopsidae): New Data on Its Distribution, Ecology and Biology"

_insects, 2023, doi:10.3390/insects14100789_

Round 1

Reviewer 1 Report

An interesting study on the distribution and life cycle of a species of the rare Prophalangopsidae written in good and understandable English. There are only some minor points that could improved.

The manuscript is generally written in good and understandable English, but there are several minor points that could improved.

Author Response

Dear reviewer,

Thank you very much for review and useful comments. I have corrected the English according to your recommendations. Changes made in manuscript are highlighted in red in the text.

Reviewer 2 Report

The manuscript ”The mysterious Amurian grig Paracyphoderris erebeus 2 Storozhenko, 1980 (Orthoptera: Prophalangopsidae): New data 3 on its distribution, ecology and biology” is a very interesting paper that gives important details about the biology, distribution and ecology of P. erebeus and also gives some consideration on its conservation status under the distribution model predictions. The paper is well structured and the content is well written. The paper is fit to be published in the Journal with some minor modifications. I have made some suggestions in the attached file.

The paper must be read by an English speaking person to correct the minor language issues.

Author Response

(The authors gave the same response as above.)
